# Analysis of Portuguese Physiotherapists’ Self-Knowledge on Temporomandibular Disorders

**DOI:** 10.3390/ijerph20021294

**Published:** 2023-01-11

**Authors:** Paula Moleirinho-Alves, Pedro Cebola, Xavier Melo, Sérgio Simões, Catarina Godinho

**Affiliations:** 1Escola Superior de Saúde Egas Moniz, Egas Moniz School of Health and Science, 2829-511 Almada, Portugal; 2Egas Moniz Physiotherapy Clinic and Research Centre, Interdisciplinary Research Centre (CiiEM), Egas Moniz School of Health and Science, 2829-511 Almada, Portugal; 3Cuf Tejo Hospital, 1300-352 Lisboa, Portugal; 4Centro Interdisciplinar de Estudo da Performance Humana (CIPER), Faculdade de Motricidade Humana Universidade de Lisboa, 1496-751 Oeiras, Portugal

**Keywords:** physiotherapy, temporomandibular disorder, education, self-knowledge, treatment

## Abstract

Background: Physiotherapy is one of the most referenced and effective conservative strategies for treating patients with temporomandibular disorders (TMD). This study aimed to characterize and analyze the self-knowledge of TMD of Portuguese physiotherapists. Methods: an online questionnaire was carried out, and the data collected were descriptively analyzed. Results: A total of 338 physiotherapists participated, of which only 142 treated patients with TMD. Seventy-six percent of the physiotherapists reported that they had not received training in the TMD area during the physiotherapy degree course. Only 11% of the physiotherapists reported that treating patients with TMD adequately identified all symptoms of TMD. Conclusions: the present study showed that it is necessary to integrate TMD-related content into the basic training of physiotherapists and promote an increase in evidence-based training.

## 1. Introduction

Temporomandibular disorder (TMD) is one pathology that affects the masticatory muscles, the temporomandibular joint (TMJ), and associated structures [1]. The presence of persistent pain in the TMJ and/or masticatory muscles is the main reason patients seek medical help [2]. Other signs and symptoms commonly manifested by patients with TMD include a limitation of mandibular movements, chewing difficulties, joint sounds (click, clicking, or crackling), pressure sensation in the TMJ and masticatory and cervical muscles, head and cervical spine pain [1,3], and otologic symptoms such as otalgia or tinnitus [4]. The variety of symptoms discloses the complexity of this disorder, which presents a large multiplicity of risk factors [5].

The worldwide prevalence is estimated to be about 10% of the adult population [6]. In the United States of America (USA), TMD is the second most prevalent musculoskeletal condition—National Institutes of Health (NICDR, 2014) [6]. It is estimated that TMD is symptomatic in 25% of adolescents, whereas in developed countries, it is estimated that this value is between 2% and 6% [7]. In adults over 45 years of age, the prevalence of painful TMD is likely to be between 2% and 7% [8]; however, estimates are quite variable, and although individuals over 45 years of age present objective signs of TMD, they tend to be asymptomatic [9]. Regardless of the symptomatology, studies suggest that the quality of life of patients with TMD is impaired [10,11,12].

The etiology of TMD is multifactorial and biopsychosocial, resulting from triggering, predisposing, and perpetuating factors, such as macro- or microtrauma, changes in healing processes, the patient’s psychosocial profile, and genetic or epigenetic factors [13]. According to Slade et al. (2016) [5], TMD is commonly the result of multiple risk factors, so no single risk factor is sufficient by itself to cause TMD.

The diagnosis and classification of TMD is currently standardized using the Diagnostic Criteria for Temporomandibular Disorders (DC-TMD), allowing important data on prevalence and incidence to be obtained [4]. Regarding the TMD subgroups, data suggest that myofascial pain disorders are the most common, with a higher prevalence in women in all subgroups, especially those of muscular origin [14].

A multidisciplinary approach is essential for the successful treatment of patients with TMD, especially in chronic manifestations of the disease. This team should include dentists, physiotherapists (PT), speech therapists, and psychologists [15]. Physiotherapy is one of the conservative interventions used in the treatment of TMD [16], aimed to control/eliminate the symptoms [17]. Several protocols have been developed and proven effective to increase the mobility of the TMJ and/or jaw muscles through controlled jaw movements, contraction–relaxation techniques, joint mobilizations, soft tissue mobilization, and stretching [16,18].

Education/training and knowledge of the physiotherapists involved in the treatment of TMD are fundamental for the success of the intervention [19]. However, this is not commonly the case, as only a small percentage has received proper training [15]. Currently, there are 19 higher education institutions in Portugal that teach the bachelor’s degree in physiotherapy. From the analysis of the content of the curricula, it is not possible to objectively determine how many curricular units and teaching hours are specifically allocated to the TMD area. Thus, it becomes relevant to characterize and analyze the self-knowledge of Portuguese physiotherapists, with intervention in the musculoskeletal area of TMD.

## 2. Materials and Methods

### 2.1. Study Design

We carried out a cross-sectional, questionnaire-based survey study of Portuguese physical therapists.

### 2.2. Sampling and Recruitment

The sample consisted of physiotherapists with intervention in the musculoskeletal area recruited via email (contact with the Physical Medicine and Rehabilitation services of several Portuguese hospitals and other reference centers) and through social networks (Facebook and Instagram). The researchers asked the physiotherapy coordinator of the Physical Medicine and Rehabilitation services at the hospitals and referral centers to make the questionnaire link available to all PT in the respective services. The inclusion criteria defined were the completion of informed consent and the PT having an area of intervention in the musculoskeletal area. Exclusion criteria were PT having an exclusive area of intervention in the remaining areas of physical therapy (e.g., lymphatic drainage, cardio-respiratory, neurology, ergonomics, etc.).

### 2.3. Questionnaire

The questionnaire was constructed based on the questionnaire carried out by Gadotti et al. (2018) [15] and in the TMD Fact Sheet of the International Association for the Study of Pain (IASP, 2017) [20]. It was composed of 20 questions related to demographic and education/training characteristics and questions related to specific knowledge of TMD (epidemiology, pathophysiology, clinical characteristics, and treatment).

### 2.4. Ethics and Procedures

The Ethical Committee of Egas Moniz University Institute approved the study protocol on 6 May 2021 (ID N° 973/2021). All individuals gave their informed consent to the Helsinki Declaration and understood that they were free to withdraw from the study at any time.

### 2.5. Data Collection

Data were collected between 1 June 2021 and 31 October 2021 through an online questionnaire published on social networks and via email.

### 2.6. Data Analysis

Descriptive statistics were calculated to analyze the responses. Data were presented as total participants (n), frequency (%), mean, standard deviation (SD), and range. Written information provided by some PT was considered and presented.

Clustered data on the knowledge of the symptoms of DTM, the factors that contribute to disc anterior displacement, acute and chronic pain in DTM, and the strategies to evaluate and intervene during the acute and chronic phases of DTM were modeled with Generalized Estimating Equations fitted on a Poisson family distribution and an exchangeable correlation structure using the geeglm function from the geepack package (version 1.3.9) [21] in R software [22]. Fixed factors were defined as sex (men and women), age (<45 years old: younger adults; ≥45 years: middle-aged adults), and hours of training (<20 h: low; ≥20 h: high), interacting with each other by design.

## 3. Results

### 3.1. Demographics and Characteristics of the Participants

Of the 19 higher health education institutions in Portugal that teach the bachelor’s degree in physiotherapy, we obtained responses from participants from 14 (74%) different institutions, of which 36% came from the northern region of the country, 57% from the central region, and 7% from the southern region.

A total of 338 physiotherapists participated in this study. The mean age of the participants was 28.3 ± 3.58 years, with an age range between 22 and 61 years, and 247 participants were women (73%). Two hundred and eighty-five participants (74%) had a bachelor’s degree. Most participants (76%) were not instructed in TMD during their degree in physiotherapy, and only 39% had continued training in the area after completion. Only 142 physiotherapists (42%) in the study assess and treat patients with TMD. Most participants (52%) who intervened with patients with TMD worked in private clinics, and only 9 physiotherapists (6%) treated patients with TMD exclusively. The remaining demographic details and characteristics of the participants (means, standard deviation, and range) can be seen in Table 1.

### 3.2. Self-Perception of the Knowledge of Physiotherapists in the Treatment of TMD

Of the 142 physiotherapists who evaluate and treat patients with TMD, only 11% correctly indicated all symptoms related to the condition. The TMD symptom that most physiotherapists reported (71%) was pain in the temporomandibular joint, followed by pain in the masticatory muscles (68%). The etiological factors mentioned as the main risk factors for TMD were malocclusion (81%), postural changes (76%), and loss of teeth (65%). Dental malocclusion (67%; 62%) and postural changes (56%; 46%) were the main risk factors for anterior disc displacement and chronic TMD pain, respectively. The answers provided are displayed in Table 2.

### 3.3. Methods Used to Evaluate and Treat TMD Patients

The parameters most frequently evaluated by physiotherapists were local pain (100%), functional limitations of the jaw (84%), and postural changes (73%). Only 16% of the participants reported assessing sleep quality and 11% the presence of headaches (Table 3).

The intervention strategies most frequently used to treat patients with TMD were manual orofacial therapy, either in the acute phase (78%) or the chronic phase (86%), followed by postural correction strategies (62% in the acute and 68% in the chronic phase). The performance of therapeutic exercises represented only 11% of the strategy used in the acute phase and 10% in the chronic phase (Table 4).

A sex by age and by training hours interaction effect was observed for the knowledge of the symptoms of DTM (X2 = 8.73, *p* = 0.0031), the factors that contribute to acute (X2 = 14.73, *p* = 0.00012) and chronic (X2 = 24.70, *p* = 0.00006) pain in DTM, the factors that contribute to disc anterior displacement (X2 = 5.13, *p* = 0.023), and the strategies to intervene during the acute (X2 = 5.07, *p* = 0.024) and chronic (X2 = 8.83, *p* = 0.003) phases of DTM. An age by training hours interaction effect (X2 = 6.92, *p* = 0.0) was observed for the knowledge of strategies to evaluate patients with DTM.

Overall, this suggests that responders with ≥20 h training, in particular young responders, correctly identified the factors that contribute to acute (men = 8.45%; women = 3.52%) and chronic pain in DTM (men = 8.5%; women = 10.52%), the factors that contribute to disc anterior displacement (men = 4.93%; women = 6.34%), and the strategies to intervene during the acute (women = 2.82%) and chronic phases of DTM (men = 17.61%; women = 19.1%) compared to others. It also suggests that young women with ≥ 20 h training (4.93%) correctly identified the symptoms of DTM and that middle-aged women with ≥20 h training (2.82%) correctly identified the strategies to intervene during the acute phase of DTM compared to others. Finally, responders with ≥20 h training, in particular young responders (13.38%), correctly identified the strategies to evaluate patients with DTM compared to others.

## 4. Discussion

This is the first study to assess the experience and knowledge of Portuguese physiotherapists regarding the treatment of patients with TMD. Our findings highlight the need for adequate training on TMD treatment in undergraduate and postgraduate physiotherapists.

The age range in the present study was lower than the Gadotti et al. (2020) [15] (average age = 45 years old), but the prevalence of female responders (73%) was consistent with the studies led by Gadotti et al. (2020) [15] (66%) and Dalanon et al. (2020) [19] (59.57%).

Only 24% of the responders reported that they had TMD educational content during the physiotherapy course, and 39% went on a continuing education course in TMD. This is important as we found that the number of training hours, especially in middle-aged physiotherapists, was a significant factor in several of the tested variables, including the knowledge of the symptoms. This suggests that a revision of the curricular contents of the physiotherapy degree is necessary, due to the increased frequency of TMD symptoms and bruxism behaviors associated with the increased stress experienced during the COVID-19 pandemic [23]. Of the 142 physiotherapists who responded that they treated TMD, only nine worked exclusively with patients with TMD, and 52% worked in private physiotherapy clinics. These percentages were similar to those found by Gadotti et al. (2020) [15] and likely resulted from the scarcity of workplaces with integrated multidisciplinary teams. Still, 38% and 31% of physiotherapists stated that their perception of the level of knowledge was satisfactory and good, respectively. These data are somewhat contrary to Gadotti et al. (2020) [15], in which 50% of physiotherapists were not confident in treating patients with TMD. This might be explained by the increased recognition gap between physiotherapist’s self-assessment and knowledge [24].

Regarding the knowledge of TMD, only 11% (of the 142 physiotherapists) correctly identified the symptoms of TMD. This value was in line with the low training on TMD in both undergraduate and postgraduate education.

About 90%, 77%, 71%, and 68% identified TMJ click and crackle, oral opening range limitation, TMJ pain, and muscle pain in masticatory muscles and accessories as symptoms of TMD, respectively. However, only 11% identified headache as a symptom of TMD, in clear contrast with the values reported by Gadotti et al. (2020) [15] (90%). Headaches and TMD often coincide and relate. The more symptoms of TMD a person holds, the more frequent their headaches are, and vice versa. Due to the biomechanical aspects of TMD and headaches, there is a constant interaction between these, as TMD can cause headaches due to pain in the masticatory muscles. We also know that physiotherapy can contribute to improving this type of headache attributed to TMD, mainly associated with aerobic physical exercise [25]. We also found that only 24% identified tinnitus and/or other ear symptoms as a symptom of TMD, which is likely explained by the lack of extensive literature on the subject. Still, Manfredini et al. (2015) [26] reported a prevalence of tinnitus in 30.4% of patients with TMD, and Buergers et al. (2014) [27] suggested that tinnitus is eight times more prevalent in patients with TMD.

Regarding etiological factors, 81%, 76%, and 65% of the responders identified malocclusion, postural changes, and missing teeth as contributors to TMD, although these are considered minor factors in the scientific literature [28]. The most recent bibliography indicates that sensitivity to stress, behavior, parafunction, and psychological are relevant factors in the etiology of TMD [29], but these factors did not have a significant expression in the responses given by Portuguese physiotherapists (29%, 25%, 30%, and 46%, respectively).

Myalgia masticatory muscles are the most common subgroup of TMD, with a prevalence of 5 to 10% [30]. However, only 14% of the physiotherapists in Portugal identified these correctly. As for anterior disc displacement, 61% of the sample responded that it could drive to conditions leading to pain in TMD, although the literature suggests that anterior disc displacement can be asymptomatic and arthralgia can be independent of changes in intra-anatomical factors [31]. We know that parafunction and (behavioral) bruxism have a strong influence on factors contributing to anterior disc displacement [32]; however, only 17% and 14% of Portuguese physiotherapists, respectively, identified these correctly. The most commonly identified factors were dental malocclusion (67%) and postural changes (56%), although these tended to be subvalued in current literature.

Regarding the evaluation of TMD, the most common parameters assessed were local pain (100%) and jaw functional limitation (84%). We consider that these are essential characteristics and integrate the axis I of evaluation of the Research Diagnostic Criteria for Temporomandibular Disorders (RDC/TMD) and the Diagnostic Criteria for Temporomandibular Disorders (DC/TMD) [33]. However, assessing the psychosocial context is increasingly relevant and contributes significantly to TMD and pain in general [34,35]. Although 56% of Portuguese physiotherapists reported depression, only a minority of Portuguese physiotherapists reported anxiety (32%) and stress (19%) as an evaluation parameter. Another parameter hardly mentioned by the Portuguese physiotherapists was sleep quality (16%). This is a growing topic in research and in daily clinical practice. It is well established that the relationship between pain and sleep quality is bidirectional; that is, poor sleep quality triggers and exacerbates pain and vice versa [5].

The most common treatment method for acute and chronic pain among Portuguese physiotherapists was oral manual therapy. While this method is effective in acute pain [36], in chronic pain, the appropriate methods for pain control, in addition to orofacial manual therapy, are exercises, education, counseling, pain catastrophizing, and fear of pain [2].

This study is not without limitations. We applied an online questionnaire that in itself has limitations. The questionnaire, although built on the basis of previous studies, was not subject to validation. Although the number of participants was large, we cannot attest its representativeness of the Portuguese reality. The results of this study should be interpreted with caution due to low external validity. In a future study, we would like to carry out a validated multicenter questionnaire in a similar time frame. It would also be essential to add illustrative images on google forms and/or even take the questionnaire in person.

## 5. Conclusions

This study highlights the need for adequate training on treating TMD and sets the stage for an open and multidisciplinary discussion on a curricular review in undergraduate and postgraduate education.

## Figures and Tables

**Table 1 ijerph-20-01294-t001:** Demographics and characteristics of the participants (total sample and sample of physiotherapists with temporomandibular disorders practice).

**Age in years** (*n* = 338) (mean, standard deviation, and range)	28.3 ± 3.58 (22–61)
(*n* = 142) (mean, standard deviation, and range)	34.1 ± 8.2 (22–61)
**Sex, female/male** (*n* = 338) (total, percentage)	247 (73%)/91 (27%)
(*n* = 142) (total, percentage)	87 (62%)/55 (38%)
**Highest level of education** (*n* = 338/*n* = 142) (total, percentage)	
Bachelor’s degree	285 (74%)/111 (78%)
Master’s degree	51 (15%)/30 (21%)
Doctoral degree	2 (1%)/1 (1%)
**Physiotherapy course with TMD educational content** (*n* = 338) (total, percentage)	
Yes	81 (24%)
No	257 (76%)
**Continuing education in TMD** (*n* = 338) (total, percentage)	
Yes	132 (39%)
No	206 (61%)
**Hours of continuing education in TMD** (*n* = 132) (total, percentage)	
Between 10 and 19 h	34 (26%)
Between 20 and 29 h	15 (11%)
More of 30 h	83 (63%)
**Evaluate or treat patients with TMD** (*n* = 338) (total, percentage)	
Yes	142 (42%)
No	196 (58%)
**Areas of intervention for participants who treat TMD** (*n* = 142) (total, percentage)	
Exclusive TMD	9 (6%)
TMD and musculoskeletal	32 (23%)
Several areas	101 (71%)
**Work setting in TMD** (*n* = 142)	
Private Physiotherapy Clinic	73 (52%)
Dental Medicine Clinic	10 (7%)
Private Hospital	9 (6%)
National Health System	11 (8%)
Private Physiotherapy Clinic + Dental Medicine Clinic	16 (11%)
Others	23 (16%)
**Years of practice in TMD** (*n* = 142)	
(−1 year)	22 (15%)
(1–5 years)	79 (56%)
(6–10 years)	36 (25%)
(+11 years)	5 (4%)
**Patients with TMD/week** (*n* = 142)	
(≤1 patient/week)	45 (32%)
(1–5 patient/week)	59 (41%)
(6–10 patient/week)	18 (13%)
(11–19 patient/week)	13 (9%)
(+20 patient/week)	7 (5%)
**Perceived level of knowledge** (*n* = 142)	
Unsatisfactory	18 (13%)
Weak	10 (7%)
Satisfactory	54 (38%)
Good	44 (31%)
Very good	16 (11%)

**Table 2 ijerph-20-01294-t002:** Self-perception of the knowledge of physiotherapists in the treatment of TMD (*n* = 142).

**TMD symptoms**	**Total**	**percentage**
Correctly identified symptoms	16	11%
Symptoms partially correctly identified	126	89%
**TMD symptoms identified**	**Total**	**percentage**
TMJ pain	101	71%
Muscle pain of masticatory muscles and accessories	97	68%
Muscle pain of other masticatory muscles	23	16%
Headache	16	11%
TMJ click and crackle	128	90%
Oral opening range limitation	109	77%
Tinnitus and/or other ear symptoms	34	24%
**Etiological Factors**	**Total**	**percentage**
Malocclusion	115	81%
Missing teeth	92	65%
Parafunctions	43	30%
Postural changes	108	76%
Stress	41	29%
Psychological	65	46%
Behavioral (example: bruxism)	36	25%
Changes in pain processing systems	9	6%
Other pain comorbidities	57	40%
**Conditions associated with pain in TMD**	**Total**	**percentage**
Myalgia masticatory muscles	20	14%
TMJ osteoarthritis	11	8%
Disc anterior displacement	61	61%
TMJ osteoarthritis + disc anterior displacement	20	14%
Myalgia masticatory muscles + disc anterior displacement	30	21%
**Factors that contribute to anterior disc displacement**	**Total**	**percentage**
Parafunctions	24	17%
Behavioral (example: bruxism)	20	14%
Postural changes	80	56%
Dental malocclusion	95	67%
Trauma	18	13%
One-sided chewing	68	48%
**Factors contributing to the chronicity of TMD**	**Total**	**percentage**
Postural changes	65	46%
Psychosocial factors	34	24%
Fear of movement	38	27%
Dental malocclusion	88	62%
Catastrophizing pain	27	19%
Self-efficacy strategies	18	13%

**Table 3 ijerph-20-01294-t003:** Parameters of TMD evaluation most often used (*n* = 142).

TMD Evaluation	Total	Percentage
Local pain	142	100%
Generalized pain	38	27%
Headache	16	11%
Catastrophizing pain	57	40%
Jaw functional limitation	120	84%
Symptoms duration time	97	68%
Postural changes	104	73%
Comorbidities	62	44%
Parafunctional habits	34	24%
Fear of movement	41	29%
Anxiety	45	32%
Stress	27	19%
Self-efficacy strategies	62	44%
Depression	80	56%
Sleep quality	23	16%

**Table 4 ijerph-20-01294-t004:** Methods used to treat patients with TMD (*n* = 142).

Methods Used in the Acute Phase		Methods Used in the Chronic Phase	
Orofacial manual therapy	111 (78%)	Orofacial manual therapy	122 (86%)
Postural correction	88 (52%)	Postural correction	97 (68%)
Education and counseling	54 (38%)	Education and counseling	471 (33%)
TENS	17 (11%)	Cervical manual therapy	241 (17%)
Exercises	16 (11%)	Exercises	14 (10%)
Ultrasound	14 (10%)	Ultrasound	11 (10%)
		TENS	5 (8%)

## Data Availability

The data presented in this study are available upon request from the first author.

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
