# Peer review of "Analysis of Portuguese Physiotherapists’ Self-Knowledge on Temporomandibular Disorders"

_ijerph, 2023, doi:10.3390/ijerph20021294_

Round 1

Reviewer 1 Report

The authors report results from a survey of Portuguese physical therapists regarding their knowledge about TMD. The paper is well written, the methods are adequate.

However, the authors should check if their data permits for some hypothesis testing, e.g. the association between level of TMD education and knowledge about symptoms. This would allow for more specific recommendations for future studies and PT training. 

Comments: 

Major:

1.) Methods, study design (L76-77, L148). In my opinion, this is not an observational study. Suggestion: We carried out a cross-sectional, questionnaire-based survey study of Portuguese physical therapists.

2.) Please specify inclusion/exclusion criteria (L87): What are “remaining areas of physical therapy” (e. g. lymphatic drainage)? In general, the use of “physical therapists in the musculoskeletal area” (L72) is awkward, since most PT work with musculoskeletal problems. Rather just specify that PT working exclusively in lymphatic drainage etc. were excluded. 

3.) Explore the possibility of further exploratory statical analysis, and discuss why it was/wasn’t performed, see above.

4.) Discussion (L233): Is the study by Gadotti et al. (ref. 17) not a similar self-assessment? 

Minor: 

5.) Data analysis (L102): Table 1 also reports means, SD and range => Please state so in this section.

6.) Section 3.1. Inconsequent writing style. Recommendation: Consequently report n and %, e.g. “only 142 PT (42%) in the study …” 

7.) Table 4: Sort either by proportion or alphabetically, not random. 

8.) The manuscript, especially the discussion section, could benefit from English proofreading for style and grammar, e. g.:

L108: Remove “Subsection”

L111: “Had a bachelors degree”

Table 1 (Patients with TMD/week): -1 patients should probably read: ≤1

Table 1/L120: “Perceived level of knowledge”

Table 2: Rephrase “Conditions lead to pain in TMD” for clarity => e. g. “Conditions associated with pain in TMD”

Author Response

The authors report results from a survey of Portuguese physical therapists regarding their knowledge about TMD. The paper is well written, the methods are adequate.

However, the authors should check if their data permits for some hypothesis testing, e.g. the association between level of TMD education and knowledge about symptoms. This would allow for more specific recommendations for future studies and PT training. 

Comments: 

Major:

1.) Methods, study design (L76-77, L148). In my opinion, this is not an observational study. Suggestion: We carried out a cross-sectional, questionnaire-based survey study of Portuguese physical therapists.

We thank the reviewer for this suggestion. We have analyzed and we accept your suggestion. Please find the changes that were made in lines 85-86.

2.) Please specify inclusion/exclusion criteria (L87): What are “remaining areas of physical therapy” (e. g. lymphatic drainage)? In general, the use of “physical therapists in the musculoskeletal area” (L72) is awkward, since most PT work with musculoskeletal problems. Rather just specify that PT working exclusively in lymphatic drainage etc. were excluded. 

We have added as an example several areas of specialization in physiotherapy in the exclusion criteria in lines 97-98.

3.) Explore the possibility of further exploratory statical analysis, and discuss why it was/wasn’t performed, see above.

We thank the reviewer for this suggestion. We have decided to model the clustered data with Generalized Estimating Equations fitted on poisson family distribution and a exchangeable correlation structure using the geeglm fuction from geepack package (version 1.3.9) in R software. Please find the statistical analysis methodology that was added in lines 117-122 and the results in lines 173-189.

4.) Discussion (L233): Is the study by Gadotti et al. (ref. 17) not a similar self-assessment? 

Acknowledging the reviewer’s question, it is important to clarify that our questionnaire is similar  to that of Gadotti et al (ref 17) carried out in Florida (USA), but ours is the first carried out in Portugal. 

Minor: 

5.) Data analysis (L102): Table 1 also reports means, SD and range => Please state so in this section.

We thank the reviewer for this suggestion, we have added the statement to line 139. 

6.) Section 3.1. Inconsequent writing style. Recommendation: Consequently report n and %, e.g. “only 142 PT (42%) in the study …” 

7.) Table 4: Sort either by proportion or alphabetically, not random. 

8.) The manuscript, especially the discussion section, could benefit from English proofreading for style and grammar, e. g.:

L108: Remove “Subsection”

L111: “Had a bachelors degree”

Table 1 (Patients with TMD/week): -1 patients should probably read: ≤1

Table 1/L120: “Perceived level of knowledge”

Table 2: Rephrase “Conditions lead to pain in TMD” for clarity => e. g. “Conditions associated with pain in TMD”

We thank the reviewer for the previous 3 suggestions. We have accepted them and made the respective changes.

Reviewer 2 Report

The topic submitted for review is very important and undoubtedly important to analyze. However, this kind of research requires much more methodological commitment. Showing n and percentage is not enough. It is worth analysing between gender, level of education, length of service. Also, it is worth making a breakdown between the age ranges of the physiotherapists surveyed. At this stage, the work does not meet the requirements.

Author Response

The topic submitted for review is very important and undoubtedly important to analyze. However, this kind of research requires much more methodological commitment. Showing n and percentage is not enough. It is worth analysing between gender, level of education, length of service. Also, it is worth making a breakdown between the age ranges of the physiotherapists surveyed. At this stage, the work does not meet the requirements.

Acknowledging the reviewer’s feedback, we have further added to the statistical analysis the following (lines 117-122):

“Clustered data was modeled with Generalized Estimating Equations fitted on poisson family distribution and a exchangeable correlation structure using the geeglm fuction from geepack package (version 1.3.9) in R software. Fixed factors were defied as sex (men and women), Age (<45 years-old: younger adults; ≥45 years: middle-aged adults), and Hours of Training (<20h: low; ≥20h high), interacting with each other by design.”

The analysis results were added to lines 173-189.

Reviewer 3 Report

Dear Authors,

I read with interest your manuscript, which highlights the finding of specific training deficiencies for Portuguese physiotherapists in relation to the treatment of Temporomandibular Disorders. I can suggest some ideas for improving the scientific paper evaluated and I have only one question regarding the training of the physical therapy specialists investigated in this study.

1. The introduction is scientifically sound. Here you clearly present the factors and symptoms associated with TMD. Perhaps it would be useful to add a short paragraph highlighting the issues related to the theoretical and practical scientific training of physiotherapists (by investigating the contents of curricula and scientific subjects in Portuguese higher education institutions). In this way you would make the connection more clearly with the research direction specified in lines 71-73.

2. Sampling and recruitment: Is the investigated lot representative for all of Portugal (the participants were selected from the main areas of your country)?

3. Some ideas/data related to validity and internal consistency for the questionnaire taken and adapted from sources 17 and 20.

4. Perhaps a construction of the questionnaire by questions using the Likert scale (1-5 or 1-7) would have facilitated more comparisons within the investigated group. I think that this variant would have allowed the use of parametric ANOVA techniques, highlighting the differences of opinion between men and women or between the different groups according to experience (-1 year, 1-5 years, 6-10 years, +11 years). It is only a suggestion, presenting the answers in percentages is a useful and very well presented option by you, but which cannot investigate these aspects as well.

5. I understand that source 17 is relevant to your study. She is quoted excessively though: lines 62,70,90,155,161,168,173,182.

6. Table 1 is obviously related to the participants, not the Results section.

7. The investigated batch has 338 participants, but only 142 physiotherapists have experience and have treated patients with TMD (it is normal that only their answers are presented in tables 2-4). Perhaps the data related to sex, age and highest level of education could be presented separately, for this defining group in the conducted study.

8. Lines 244-248 argue the importance of your research. It would be useful to identify some concrete solutions to the identified problems (presentation of the results of university teaching staff, changes to the curriculum at the university level, the inclusion of other scientific subjects of study, the increase in the number of hours allocated for training related to TMD..............

The question: Did the questioned physiotherapists graduate from bachelor's/master's/doctorate university programs within the Faculties of Physical Education and Sport or within some specializations belonging to the Faculties of Medicine?

Author Response

I read with interest your manuscript, which highlights the finding of specific training deficiencies for Portuguese physiotherapists in relation to the treatment of Temporomandibular Disorders. I can suggest some ideas for improving the scientific paper evaluated and I have only one question regarding the training of the physical therapy specialists investigated in this study.

1.The introduction is scientifically sound. Here you clearly present the factors and symptoms associated with TMD. Perhaps it would be useful to add a short paragraph highlighting the issues related to the theoretical and practical scientific training of physiotherapists (by investigating the contents of curricula and scientific subjects in Portuguese higher education institutions). In this way you would make the connection more clearly with the research direction specified in lines 71-73.

Thank you for the reviewer’s feedback. Currently there are 19 higher education institutions in Portugal that teach the bachelor’s degree in physiotherapy. From the analysis of the contents of curricula it ‘isn’t possible to objectively determine how many curricular units and teaching hours are specifically allocated to the TMD area. Thus, it becomes relevant in this study line 75-79.

  1. Sampling and recruitment: Is the investigated lot representative for all of Portugal (the participants were selected from the main areas of your country)?

Thank you for the reviewer’s question. From the 19 higher education institutions in Portugal that teach the bachelor’s degree in physiotherapy we obtained responses from participants from 14 (74%) different institutions, of which 36% from the northern region of the country, 57% from the central region and 7% from the southern region (added to the results section lines 126-129). Furthermore, we have included a related statement in the discussion section (lines 197-198).

  1. Some ideas/data related to validity and internal consistency for the questionnaire taken and adapted from sources 17 and 20.

Thank you for the reviewer’s suggestion. The questionnaires constructed in the reference studies 17 and 20 were also not validated. However, as we recognize that this may constitute a problem with the validity of our study, we introduce as a limitation to our study the fact that our questionnaire has not been validated line 299-301.

  1. Perhaps a construction of the questionnaire by questions using the Likert scale (1-5 or 1-7) would have facilitated more comparisons within the investigated group. I think that this variant would have allowed the use of parametric ANOVA techniques, highlighting the differences of opinion between men and women or between the different groups according to experience (-1 year, 1-5 years, 6-10 years, +11 years). It is only a suggestion, presenting the answers in percentages is a useful and very well presented option by you, but which cannot investigate these aspects as well.

Acknowledging the reviewer’s suggestion, we have further added to the statistical analysis the following (lines 117-122):

“Clustered data was modeled with Generalized Estimating Equations fitted on poisson family distribution and a exchangeable correlation structure using the geeglm fuction from geepack package (version 1.3.9) in R software. Fixed factors were defied as sex (men and women), Age (<45 years-old: younger adults; ≥45 years: middle-aged adults), and Hours of Training (<20h: low; ≥20h high), interacting with each other by design”.

The analysis results were added to lines 173-189.

  1. I understand that source 17 is relevant to your study. She is quoted excessively though: lines 62,70,90,155,161,168,173,182.

We understand and agree with the reviewer’s comment but unfortunately this is the only reference study regarding this subject that is applicable to our work.

  1. Table 1 is obviously related to the participants, not the Results section.

We understand the reviewer’s comment, nevertheless since our study didn’t had a predetermined sample, it was not possible to know the sample’s characteristics beforehand making this information, on our opinion, more suitable to be included in the results section. 

  1. The investigated batch has 338 participants, but only 142 physiotherapists have experience and have treated patients with TMD (it is normal that only their answers are presented in tables 2-4). Perhaps the data related to sex, age and highest level of education could be presented separately, for this defining group in the conducted study.

We accept the reviewer’s suggestion and have made corresponding changes to table 1. Additionally we have included the age average, SD, sex and level of education of the 142 participants that intervene in the TMD area. 

  1. Lines 244-248 argue the importance of your research. It would be useful to identify some concrete solutions to the identified problems (presentation of the results of university teaching staff, changes to the curriculum at the university level, the inclusion of other scientific subjects of study, the increase in the number of hours allocated for training related to TMD..............

We understand the reviewer’s suggestion, therefore we have included in the discussion section (lines 291-294) the suggestions that in our opinion could be followed to put in practice the outcomes of our study.

The question: Did the questioned physiotherapists graduate from bachelor's/master's/doctorate university programs within the Faculties of Physical Education and Sport or within some specializations belonging to the Faculties of Medicine?

Acknowledging the reviewer’s question, it is important to clarify that in Portugal, the physiotherapy course is taught in higher health schools, which are independent from sports and medicine schools. Hoping to make this clearer, we have added further clarification on lines 75-76.

Round 2

Reviewer 1 Report

Thank you for providing a revised version. The previous major comments were sufficiently addressed, however I still recommend minor revisions before publication: 

1.) Please include in the methods section (L117) what the chosen outcome variables were (e.g. knowledge of symptoms). In the respective results section, it should probably say “TMD”. The discussion section should have a dedicated paragraph for the regression analyses, summarizing the results and their implication. Lines 288-290 are a starting point for this paragraph.

2.) In the limitations section, it still says “We would also like to conduct a statistical analysis between the type of answer and the degree (bachelor's, licentiate, master's, and doctorate).” However, given that the authors have now chosen to calculate multivariate statistical models for some of the associations in question, it remains unclear why the degree was not included if these analyses if it is a desired variable. 

3.) Table 1: The age and level of education is now included as an additional subsection for the 142 PT with TMD experience, presumably, since it is not clearly labeled. The way Table 1 is currently presented is confusing and should be improved.

4.) I recommend a final round of English scientific proofreading for style and grammar before publication, e. g. L197-199, L255, L291, L310

5.) The authors should consider providing the questionnaire as supplementary data.  

Author Response

Thank you for providing a revised version. The previous major comments were sufficiently addressed, however I still recommend minor revisions before publication: 

1.) Please include in the methods section (L117) what the chosen outcome variables were (e.g. knowledge of symptoms). In the respective results section, it should probably say “TMD”. The discussion section should have a dedicated paragraph for the regression analyses, summarizing the results and their implication. Lines 288-290 are a starting point for this paragraph.

We thank the reviewer for this suggestion. Regarding the outcome variables we have made the following changes:

  • Added to lines124-126 in the “Materials and Methods” section.
  • Removed lines 288-290 and added the removed information to lines 221-225 of the “Discussion” section.

2.) In the limitations section, it still says “We would also like to conduct a statistical analysis between the type of answer and the degree (bachelor's, licentiate, master's, and doctorate).” However, given that the authors have now chosen to calculate multivariate statistical models for some of the associations in question, it remains unclear why the degree was not included if these analyses if it is a desired variable. 

Acknowledging the reviewer’s question, in order to clarify, we have deleted that sentence as the number of hours of training, and not the academic degree, was a significant factor in the model.

3.) Table 1: The age and level of education is now included as an additional subsection for the 142 PT with TMD experience, presumably, since it is not clearly labeled. The way Table 1 is currently presented is confusing and should be improved.

Acknowledging the reviewer’s question, in order to clarify, we have reorganized the table and its title.

4.) I recommend a final round of English scientific proofreading for style and grammar before publication, e. g. L197-199, L255, L291, L310

We thank the reviewer for this suggestion. We have analyzed it and we accept your suggestion therefore; we have performed a final style and grammar English scientific proofreading.

5.) The authors should consider providing the questionnaire as supplementary data.  

Acknowledging the reviewer’s suggestion, it is important to clarify that our questionnaire was not included in the submission because it is written in Portuguese therefore, we think that is not suitable for an international publication.

Reviewer 2 Report

In my opinion, this version of the article is definitely better.

Author Response

We would like to thank the reviewer for their feedback and support in improving this study.